# Dynamics of labor and capital in AI vs. non-AI industries: A two-industry model analysis

**Xu Huang** [ORCID]*

School of Finance and Information, Ningbo University of Finance and Economics, Ningbo, China

* huangxu@nbufe.edu.cn

## Abstract

There is an imbalance in the development of artificial intelligence between industries. Compared to non-AI enterprise, AI- enterprise will save labor, enhance innovation capabilities, and improve production efficiency. By constructing a two-industry model of AI and non-AI enterprise, this paper finds that with the development of artificial intelligence in the same industry, the AI enterprise will occupy a dominant position, attracting labor and capital from the non-AI enterprise into the AI enterprise. In different industries, the development of artificial intelligence improves the production efficiency of the enterprise. However, due to the price effect, non-AI enterprise benefits more. Labor and capital flow from AI enterprise to non-AI enterprise. In order to promote the improvement of production efficiency in the whole society, the government can tax non-AI enterprise and subsidize them to AI enterprise. Taxation promotes the degree of automation and the improvement of production efficiency, but it has only a short-term effect on the development of AI. At the same time, taxation inhibits the development of non-AI enterprise, and there is a high risk of unemployment. When both industries use artificial intelligence for production, the labor share and the capital share of the two industries will tend to the same value. The convergence of technology measures is conducive to increasing labor income share and reducing income inequality, but it is not conducive to innovation.

## Introduction

Artificial intelligence (AI) has had a significant impact on enterprise transformation and upgrading [1, 2]. AI enables automation of repetitive and mundane tasks, freeing up employees to focus on more strategic and value-added activities. By automating processes, AI improves efficiency, reduces errors, and lowers operational costs [3, 4]. AI technology facilitates innovation by enabling businesses to extract insights from data, identify customer needs, and develop new products and services. AI can assist in product design, customization, and predictive maintenance, leading to improved offerings and competitive advantage [5]. AI impacts the workforce by augmenting human capabilities and transforming job roles. While some routine tasks may be automated, AI also creates new job opportunities that require skills in managing AI systems, analyzing complex data, and developing AI applications. Adopting AI technologies can provide a competitive edge to enterprises. Businesses that leverage AI

**Data Availability Statement:** All relevant data are within the paper and its Supporting information files.

**Funding:** This research was made possible through the generous support of the Chinese National Funding of Social Sciences under Grant

No. 23BJY134, specifically contributing to the study design. We express our sincere gratitude for this support. The roles of the authors are detailed in the 'Author Contributions' section.

**Competing interests:** The authors have declared that no competing interests exist.

effectively can deliver faster and more accurate services, make data-driven decisions, and adapt to market dynamics swiftly, positioning themselves ahead of their competitors [6, 7].

In China, the development of AI applications shows regional and industry disparities [8, 9]. Major cities like Beijing, Shanghai, and Shenzhen lead in AI advancement due to resources and technology advantages, while smaller cities and rural areas lag behind due to tech, talent, and funding challenges. AI adoption also varies by industry. Finance, e-commerce, and the internet rapidly embrace AI with substantial data and digital foundations, boosting efficiency and innovation. However, traditional sectors like manufacturing progress more slowly, facing transformation challenges [10].

These differences in AI adoption leads to disparities in production efficiency and innovation, favoring AI-advanced industries [11]. Labor-intensive sectors are slower to adopt AI, risking missed efficiency and innovation gains [12]. Unequal AI deployment affects the labor market. Some industries automate tasks, reducing job opportunities. Yet, AI creates new roles in AI development and data science, often requiring advanced skills and education, potentially worsening skill disparities [13]. These disparities can exacerbate social inequality, favoring those in automated industries with better economic opportunities. Conversely, workers in low-skilled or traditional sectors may face unemployment or stagnant wages, widening wealth and opportunity gaps, especially in AI-intensive industries [14].

Accurate prediction of the impact of AI development on industrial structure transformation and upgrade is of important practical significance for promoting the high-quality development of the world economy [15]. In the same industry, some enterprises have used artificial intelligence for production, and some enterprises still use traditional technology for production. What impact will AI have on the enterprises mentioned above? Among different industries, some industries are highly intelligent, such as the automobile industry. Some industries rarely use AI for production and how AI will affect different industries. This paper constructs a theoretical model to try to answer the above questions.

In the process of leading business transformation and upgrading, artificial intelligence encompasses multiple pathways, providing profound impetus for reshaping and upgrading industrial landscapes [16, 17]. Firstly, viewing artificial intelligence as a means of automating production, by introducing more cost-effective capital to replace traditional routine tasks, enterprises have significantly improved production efficiency through the deployment of large-scale automation equipment, thereby creating a pressing demand for skilled labor [18, 19]. Secondly, artificial intelligence is interpreted as technology that can lean either toward capital or labor, depending on the elasticity of substitutability between intelligent capital and labor, subsequently influencing the selection and configuration of technological elements in the production process [20]. This variability in technological attributes drives industries toward greater efficiency and innovation capabilities. Thirdly, artificial intelligence has profound implications for market structures. Enterprises incorporating this technology gain a competitive edge by enhancing production efficiency, gradually capturing market shares, and even evolving into platform-based companies, thereby exerting dominant influence within their industries [21, 22]. This new market landscape not only enhances the competitiveness of individual enterprises but also brings transformative forces to the entire industry. Lastly, the impact of artificial intelligence on traditional production methods has led to the reallocation of capital and labor across different sectors. This flow of resources not only propels adjustments in industrial structures but also creates favorable conditions for the emergence of innovation and emerging industries [23, 24].

The development of AI will have a significant impact on the labor market [25, 26]. In terms of employment, existing literature generally suggests that AI will replace low-skilled labor while complementing high-skilled labor [27, 28]. However, AI systems like ChatGPT are

gradually replacing high-skilled labor, such as accountants and lawyers [29, 30]. AI, through its strong substitutive effect, saves labor and enhances corporate production efficiency. At the same time, AI, through its generative effect, creates new job opportunities [31]. In the long term, AI's overall impact on labor employment is relatively modest. However, in the short term, AI will lead to a dramatic transformation of the labor market. This is highlighted by the fact that displaced labor will transition from manufacturing to the service sector. AI elevates the skill levels of workers through both substitution and creation effects [32, 33].

Regarding income, existing literature generally suggests that AI will widen the wage gap between high-skilled and low-skilled labor [34, 35]. AI will promote an increase in the share of capital in the production process, leading to higher returns on capital and exacerbating income inequality. Berg et al. [36] views intelligent robots as capital that can replace labor, and since capital distribution is inherently unequal, the introduction of robots increases the share of capital and thus worsens income inequality. Korinek & Stiglitz [37] argue that AI technological innovation will intensify income distribution inequality through two channels: increasing surplus for innovators and raising the relative prices of capital factors. Acemoglu & Restrepo [38], based on their study of changes in wage structures in the United States over the past 40 years, find that the widening income gap in the United States is primarily due to declining wages in routine positions. Automation technology enhances the labor productivity of routine work tasks, thereby reducing the industry's labor income share.

Artificial intelligence is different from previous technological changes and has a greater impact on the economy [39, 40]. There are two ways to study artificial intelligence. One is to improve the neoclassical production function [41, 42]. The other is a task-based model. Acemoglu & Restrepo [43] construct a single industry model to study the impact of AI on labor, wages, and employment needs. On the one hand, automation creates an alternative effect that reduces the need for labor and wages. On the other hand, automation produces four effects to counteract substitution effects: productivity effects, capital accumulation, deepening of automation, and new tasks, with new tasks having the greatest impact. Based on a task-based model, artificial intelligence leads to an increase in the proportion of some industries, while others decline [44–46].

Based on the aforementioned analysis, this paper constructs a two-industry model to investigate the impact of artificial intelligence on enterprise transformation and upgrading. Differentiating enterprises based on their levels of intelligence, the paper categorizes them into intelligent and non-intelligent enterprises to compare the effects of artificial intelligence on enterprises with varying degrees of smartness. Existing literature has often demonstrated enterprise transformation and upgrading through the reallocation of production factors, such as labor and capital mobility. Building upon this foundation, this paper introduces the research and development (R&D) value of enterprises to explore the impact of artificial intelligence's substitution and creation effects on enterprise R&D value. This examination further stimulates the mobility of production factors, reflecting the process of enterprise transformation and upgrading.

In order to promote social development, the government can impose a certain amount of correction tax on non-AI enterprise and subsidize AI enterprise to purchase intelligent equipment. The subsidy policy will lead to the improvement of AI in the whole society, but the development of non-AI enterprise will be restrained. Restricted by the consumption effect, the development speed of AI enterprise has also slowed down, resulting in only short-term effects of the subsidy policy.

Finally, this paper discusses the model of intelligent production in both industries. Assuming that the two industries have different levels of intelligence, the share of capital and labor is different. With the progress of AI, the capital and labor of the two industries will reverse flow and the share of capital and labor of the two industries will tend to be the same in the long run.

The possible innovations of this article are as follows: Firstly, this article expands the theoretical framework of the task-based model, expanding the theoretical research of artificial intelligence from a single department to two departments. This article points out the impact of artificial intelligence on capital and labor mobility in the same industry and different industries, thereby clarifying the impact of artificial intelligence on worker skills. Secondly, this study points out the short-term and long-term effects of government subsidy policies on artificial intelligence, which has reference significance for the government to implement public policies on artificial intelligence. Thirdly, this study points out that if artificial intelligence is widely applied throughout the industry in the future, technological measures will bring changes in capital and labor shares. This has important reference significance for the future development of artificial intelligence.

## Materials and methods

The Task-based model will be employed to analyze the impact of artificial intelligence on industrial transformation and upgrading. Subsequently, this paper will utilize data from China to conduct numerical simulations aimed at validating the propositions.

### Basic assumptions

The entire society is categorized into the primary, secondary, and tertiary sectors, with the share of the primary sector increasing in China, the share of the secondary sector remaining stable, and the share of the tertiary sector continuously rising. This article does not consider the primary sector and focuses solely on the secondary and tertiary sectors. Therefore, this article employs a two-industry model for theoretical analysis.

The final products of society are produced by two industries. The production function is

$$Y_t = [\gamma Y_{1,t}^{\frac{\varepsilon-1}{\varepsilon}} + (1-\gamma) Y_{2,t}^{\frac{\varepsilon-1}{\varepsilon}}]^{\frac{\varepsilon}{\varepsilon-1}}, \tag{1}$$

where $Y_t$ is the total social output, $Y_{1,t}$ is the output of the AI enterprise, and $Y_{2,t}$ is the output of the non-AI enterprise. The AI enterprise is one that extensively utilizes AI technologies and automation equipment in its production and operations, such as a manufacturing enterprise employing industrial robots. Conversely, the non-AI enterprise is one that relies on traditional material production methods and is typically a labor-intensive enterprise. $\gamma$ is the proportion of the output of the AI enterprise among the total social output, and $\varepsilon$ is the elasticity of substitution production between the AI enterprise and the non-AI enterprise. To simplify our analysis, we assume that there is only one enterprise in each industry. If $\varepsilon > 1$, two enterprises belong to the same industry. For example, both are in the automotive industry. The AI enterprise has been automated and has adopted a machine substitution strategy that requires fewer workers. The non-AI enterprise uses traditional production equipment and requires a large amount of labor to assemble a product. If $0 < \varepsilon < 1$, the AI enterprise and the non-AI enterprise belong to different industries. For example, the AI enterprise is a manufacturing enterprise, while the non-AI enterprise is a service-oriented enterprise.

The final product price is standardized to 1. The product price produced by the AI enterprise 1 is $p_{1,t}$, and the product price produced by the non-AI enterprise is $p_{2,t}$, so that

$$p_{1,t} = \frac{\partial Y_t}{\partial Y_{1,t}} = \gamma(\frac{Y_{1,t}}{Y_t})^{-\frac{1}{\varepsilon}}, p_{2,t} = \frac{\partial Y_t}{\partial Y_{2,t}} = \gamma(\frac{Y_{2,t}}{Y_t})^{-\frac{1}{\varepsilon}} \tag{2}$$

The relative price $p_t$ is defined as

$$p_t \equiv \frac{p_{1,t}}{p_{2,t}} = \frac{\partial Y_t / \partial Y_{1,t}}{\partial Y_t / \partial Y_{2,t}} = \frac{\gamma}{1-\gamma} \cdot \left(\frac{Y_{1,t}}{Y_{2,t}}\right)^{-\frac{1}{\varepsilon}}, \tag{3}$$

and

$$[\gamma^\varepsilon p_{1,t}^{1-\varepsilon} + (1-\gamma)p_{2,t}^{1-\varepsilon}]^{\frac{1}{1-\varepsilon}} = 1. \tag{4}$$

According to Eq (3), this article represents the price effect through variations in $p_t$ When $p_t$ increases, the price of products from the AI enterprise rises more rapidly compared to that of the non-AI enterprise. The increase in product price by the AI enterprise stimulates an expansion in its production. Conversely, when $p_t$ decreases, the product price for the non-AI enterprise increases at a faster rate, prompting the non-AI enterprise to increase its output [47].

## Enterprises in two industries

Blockchain technology has several significant effects on data sharing. Blockchain is a decentralized distributed ledger where data is maintained and verified by multiple participants.

This article starts with a simple task-based model introduced by Acemoglu and Restrepo [43]. The production task in AI enterprise is $i \in [N_t - 1, N_t]$, where $N_t$ represents the new task created by AI, and also represents the creation effect of AI. The production function is

$$ln\, Y_{1,t} = \int_{N_t-1}^{N_t} ln\, y_{1,t}(i)di, \tag{5}$$

where $y_{1,t}(i)$ is the output of task $i$. Task $i$ can be produced by human labor $l_{1,t}(i)$, or by machines capital $k_{1,t}(i)$, depending on whether the task has been automated or not. If task $i \in [N_t - 1, I_t]$ has been automated, it can either be produced by workers or machines. If task $i \in (I_t, N_t]$ has not been automated, it can only be produced by workers. The threshold $I_t$ denotes the frontier of automation possibilities, indicating that AI can replace the scope of labor tasks. The threshold $I_t$ also represents the substitution effect of AI. Thus,

$$y_{1,t}(i) = \begin{cases} \gamma_{l_1}(i)l_{1,t}(i) + \gamma_{k_1}(i)k_{1,t}(i) & i \in [N_t - 1, I_t] \\ \gamma_{l_1}(i)l_{1,t}(i) & i \in (I_t, N_t] \end{cases}, \tag{6}$$

where $\gamma_{l_1}(i)$ is the productivity of labor in task $i$ and $\gamma_{k_1}(i)$ is the productivity of automated machines in task $i$. Assume that $\gamma_{l_1}(i)/\gamma_{k_1}(i)$ is monotonically increasing with respect to task $i$, it indicates that labor has a comparative advantage in high-index tasks.

The wage of the AI enterprise worker is $W_{1,t}$, and the cost of automated equipment (or the rental price) is $R_{1,t}$. To simplify our analysis, we use the following assumptions:

$$\frac{\gamma_{l_1}(I_t)}{\gamma_{k_1}(I_t)} < \frac{W_{1,t}}{R_{1,t}} < \frac{\gamma_{l_1}(N_t)}{\gamma_{k_1}(N_t - 1)}, \tag{7}$$

where the first inequality means that all tasks in $[N_t - 1, I_t]$ are produced by the automated machines, and the second inequality indicates that the introduction of new tasks will increase the output of the AI enterprise. From Eqs (5), (6) and (7), the production in the AI enterprise can be obtained as follows.

$$Y_{1,t} = A_{1,t}K_{1,t}^{I_t-N_t+1}L_{1,t}^{N_t-I_t}, \tag{8}$$

Where $A_{1,t} = \frac{\exp(\int_{N_t-1}^{I_t} \ln\gamma_{k_1}(i)di + \int_{I_t}^{N_t} \ln\gamma_{l_1}(i)di)}{(I_t-N_t+1)^{I_t-N_t+1}(N_t-I_t)^{N_t-I_t}}$ is the overall automated level of Enterprise 1. The capital used by the AI enterprise is $K_{1,t} = \int_{N_t-1}^{I_t} k_{1,t}(i)di$, and the labor force is $L_{1,t} = \int_{I_t}^{N_t} l_{1,t}(i)di$. The capital share of automated production is $I_t - N_t + 1$ and the labor share is $N_t - I_t$. Since the AI enterprise adopts a machine substitution strategy that labor share decreases with the increase of the capital share, the labor share should be less than the capital share that $0 < N_t - I_t < 1/2$.

To obtain equilibrium, the AI enterprise chooses the profit maximization strategy. The profit function is as follows.

$$\max_{K_{1,t}, L_{1,t}} \pi_{1,t} = p_{1,t}Y_{1,t} - W_{1,t}L_{1,t} - R_{1,t}K_{1,t}. \tag{9}$$

So, the first-order conditions are as follows.

$$W_{1,t} = (N_t - I_t)p_{1,t}Y_{1,t}/L_{1,t}, \tag{10}$$

$$R_{1,t} = (I_t - N_t + 1)p_{1,t}Y_{1,t}/K_{1,t}, \tag{11}$$

The following formulas can be deduced from (7), (8), (10) and (11), so that

$$\frac{\partial(\ln Y_{1,t})}{\partial I_t} = \ln\left(\frac{W_{1,t}}{\gamma_{l_1}(I_t)}\right) - \ln\left(\frac{R_{1,t}}{\gamma_{k_1}(I_t)}\right) > 0, \tag{12}$$

$$\frac{\partial(\ln Y_{1,t})}{\partial N_t} = \ln\left(\frac{R_{1,t}}{\gamma_{k_1}(N_t - 1)}\right) - \ln\left(\frac{W_{1,t}}{\gamma_{l_1}(N_t)}\right) > 0. \tag{13}$$

Eqs (12) and (13) imply that the deepening of automation and the creation of new tasks will increase the output of the AI enterprise. In the history, technological improvements increase the productivity of capital in already automated tasks. For example, a tractor that replaces motorized reapers increases productivity in agricultural production.

The non-AI enterprise adopts the traditional production method. The production function is Cobb-Douglas, so

$$\ln Y_{2,t} = \int_{M_t-1}^{M_t} \ln y_{2,t}(i)di, \tag{14}$$

where $i \in [M_t - 1, M_t]$ is the production task interval of the non-AI enterprise, task $i$ is produced by traditional capital and labor, and

$$y_{2,t}(i) = (\gamma_{k_2}(i)k_{2,t}(i))^\alpha(\gamma_{l_2}(i)l_{2,t}(i))^{1-\alpha}, \tag{15}$$

where $\gamma_{l_2}(i)$ is the productivity of labor in task $i$, and $\gamma_{k_2}(i)$ is the productivity of traditional machines in task $i$. The traditional capital used by firm 2 is $K_{2,t} = \int_{M_t-1}^{M_t} k_{2,t}(i)di$ and the labor force is $L_{2,t} = \int_{M_t}^{M_t-1} l_{2,t}(i)di$, thereby,

$$Y_{2,t} = A_{2,t}K_{2,t}^\alpha L_{2,t}^{1-\alpha}, \tag{16}$$

where $A_{2,t} = \int_{M_t-1}^{M_t} \gamma_{k_2}^\alpha\gamma_{l_2}^{1-\alpha}di$ is the technical level of the non-AI enterprise. Workers' wages in the non-AI enterprise are $W_{2,t}$, and the costs for traditional equipment (or the rental prices) are $R_{2,t}$. The AI enterprise adopts an automation strategy with a higher capital share than the non-AI enterprise, then $I_t - N_t + 1 > \alpha$.

The non-AI enterprise pursues profit maximization strategy, thereby

$$W_{2,t} = (1 - \alpha)p_{2,t}Y_{2,t}/L_{2,t}, R_{2,t} = \alpha p_{2,t}Y_{2,t}/K_{2,t}, \quad (17)$$

Assuming that the total labor force and the total capital remain unchanged when the labor market and capital market are cleared, we have

$$L_t = L_{1,t} + L_{2,t}, K_t = K_{1,t} + K_{2,t}. \quad (18)$$

Assuming that enterprises represent all laborers, the income of laborers can be divided into two components: wages and capital returns. Laborer's income is divided into two portions, one for consumption and the other for savings. Let's assume that the personal savings rate is a constant, denoted as 's'. Enterprise investments are financed through individual savings, leading to changes in the capital stock that satisfy the following equation:

$$K_{1,t+1} = (1 - \delta)K_{1,t} + I_{1,t}, K_{2,t+1} = (1 - \delta)K_{2,t} + I_{2,t}, \quad (19)$$

$$I_{1,t} = s\left(W_{1,t}L_{1,t} + R_{1,t}K_{1,t}\right), I_{2,t} = s\left(W_{2,t}L_{2,t} + R_{2,t}K_{2,t}\right). \quad (20)$$

Where $\delta$ represents the capital depreciation rate, and $I_{1,t}$ and $I_{2,t}$ denote the capital stock of enterprise 1 and enterprise 2, respectively.

## Intermediate products

The intermediate product refers to the machine and equipment used by enterprises that is produced by a monopoly enterprise. The cost of the automation machine in the AI enterprise 1 is $\varphi_{1,t}$. The enterprise producing intermediate products seeks to maximize profits, and the profit function is

$$\max_{K_{1,t}} \pi_{k_{1,t}} = (R_{1,t} - \varphi_{1,t})K_{1,t}. \quad (21)$$

Substituting (11) into (21), the following equation is obtained by first-order conditions.

$$K_{1,t} = \frac{(I_t - N_t + 1)^2 p_{1,t}Y_{1,t}}{\varphi_{1,t}}, \pi_{k_{1,t}} = (N_t - I_t)(I_t - N_t + 1)p_{1,t}Y_{1,t}. \quad (22)$$

In order to encourage innovation, enterprises that produce the intermediate products of machines and equipment for automated production always get the monopoly right to receive the benefits. The present value of the R&D the intermediate products (machines for automated production) is as follows.

$$V_{1,t} = \sum_{i=t}^{\infty} \pi_{k_{1,i}} e^{-(i-t)r_t}, \quad (23)$$

where $r_t$ is the interest rate that is assumed to be constant. From Eq (23), the HJB equation can be obtained by taking the derivatives of time.

$$r_t = \frac{\pi_{k_{1,t}}}{V_{1,t}} + \frac{V_{1,t+1}}{V_{1,t}} - 1. \quad (24)$$

Machines for traditional production in the non-AI enterprise are produced by other

enterprises that produced intermediate products. Similar to the analysis above, then

$$K_{2,t} = \frac{\alpha^2 p_{2,t} Y_{2,t}}{\varphi_{2,t}}, \pi_{k_{2,t}} = \alpha(1-\alpha)p_{2,t}Y_{2,t} \qquad (25)$$

$$V_{2,t} = \sum_{i=t}^{\infty} \pi_{k_{2,i}} e^{-(i-t)r(t)}, r_t = \frac{\pi_{k_{1,t}}}{V_{2,t}} + \frac{V_{2,t+1}}{V_{2,t}} - 1. \qquad (26)$$

## Equilibrium

This subsection analyzes the impact of using artificial intelligence for production on industries, including worker wages, R&D investment, etc.

According to (10) and (17), the relative salary $\omega_t$ is

$$\omega_t \equiv \frac{W_{1,t}}{W_{2,t}} = \frac{N_t - I_t}{1-\alpha} \cdot \frac{\gamma}{1-\gamma} \left(\frac{Y_{1,t}}{Y_{2,t}}\right)^{\frac{\varepsilon-1}{\varepsilon}} \frac{L_{2,t}}{L_{1,t}}, \qquad (27)$$

Under the equilibrium $V_{1,t+1} = V_{1,t} = V_1$, $V_{2,t+1} = V_{2,t} = V_2$, thus

$$V_1 = \pi_{k_{1,t}}/r_t, V_2 = \pi_{k_{2,t}}/r_t, \qquad (28)$$

Therefore, the ratio between the relative value of the R&D of the machines for automated production and the R&D of the machines for traditional production is as follows.

$$v_t \equiv \frac{V_1}{V_2} = \frac{\pi_{k_{1,t}}}{\pi_{k_{2,t}}} = \frac{(N_t - I)(I_t - N_t + 1)}{\alpha(1-\alpha)} \cdot \frac{\gamma}{1-\gamma} \left(\frac{Y_{1,t}}{Y_{2,t}}\right)^{\frac{\varepsilon-1}{\varepsilon}}. \qquad (29)$$

In static equilibrium, the relative wages $\omega_t$, the relative value $v_t$, the labor share and the capital share are affected by the substitution effects, the production effects and the substitution elasticity.

**Proposition 1.**

(i) When $\varepsilon > 1$, the AI enterprise and the non-AI enterprise are in the same industry. As the degree of automation deepens ($I_t$ becomes larger), the relative wages $\omega_t$ and the relative value $v_t$ of the R&D for the two machines are affected by the negative substitution effects and the positive production effects, and the overall impacts are uncertain. As the number of the new tasks increases ($N_t$ becomes larger), the relative wage $\omega_t$ and the relative value $v_t$ of the R&D investments are increased. When the change in the deepening of automation and the increase of the new tasks are the same ($dI_t = dN_t$), the impact on the relative wage $\omega_t$ and the relative value $v_t$ of the R&D investment are both positive. The deepening of automation and the increase in new tasks will increase production efficiency, increasing the output of the AI enterprise. If the output of the AI enterprise increases, the output of the non-AI enterprise decreases.

(ii) When $0 < \varepsilon < 1$, the AI enterprise and the non-AI enterprise are in different industries. As the degree of automation deepens ($I_t$ becomes larger), the relative wage $\omega_t$ and the relative value $v_t$ of R&D investment will decrease. As the number of new tasks increases ($N_t$ becomes larger), changes are uncertain for the relative wages $\omega_t$ and the relative value $v_t$ of R&D investment. When changes in automation deepening and the increase in the new task are the same ($dI_t = dN_t$), the overall impact on the relative wage $\omega_t$ and the relative value $v_t$ of R&D investment $v_t$ are negative. The deepening of automation and the increase of new tasks increase the production efficiency of the AI enterprise and increase the output of the

AI enterprise. However, the output share of the enterprise decreases and the output share of the non-AI enterprise increases.

(All proofs for Proposition 1 are contained in the S1 Appendix.).

In the same industry, the wages of workers in AI enterprise increase faster, and the R&D investments in automation equipment receive larger benefits than traditional equipment. As AI improves production efficiency, the AI enterprise occupy more market share. In peer competition, enterprises using AI accelerate the process of intelligent production. Encouraged by the wage effect and R&D profit, more enterprises will use AI for production, which will lead to an improvement in the intelligence level of the whole industry. For example, in the automobile industry, after a few enterprises purchase intelligent equipment for production, other enterprises also adopt the same strategy. The situation of mutual competition has led to an improvement of the overall intelligence level of the automotive industry, which is significantly higher than that of other industries. This also explains the emergence of superstar enterprises. Enterprises that have achieved intelligent production will attract more funds and talents to join, occupy more market share, and form a monopoly situation due to their technological advantages.

In different industries, the products produced by the two enterprises complement each other. It is assumed that the AI enterprise is a manufacturing enterprise, and the non-AI enterprise is a service enterprise. The use of artificial intelligence in the manufacturing industry has improved the production efficiency and its output will increase, thus the price of manufacturing products will fall. However, the social demand for service products is constant, so the price of service products becomes more expensive. In the end, the share of manufacturing output fell and the share of service industry output increased.

Suppose that labor and capital can flow freely between the two industries. In the equilibrium, the wages of the two industries will be equal and the gains of capital in the two industries will be the same, that is, the capital has no arbitrage.

$$W_{1,t} = W_{2,t}, R_{1,t} = R_{2,t}. \tag{30}$$

To investigate the issue of labor and capital mobility, this article assumes that the allocation of capital and labor in enterprise 1 is governed by the following function:

$$x_{k,t} = \frac{K_{1,t}}{K_t}, \; x_{l,t} = \frac{L_{1,t}}{L_t}. \tag{31}$$

Hence, the capital and labor proportions in enterprise 2 are $1 - x_{k,t}$ and $1 - x_{l,t}$ respectively. When $x_{k,t}$ and $x_{l,t}$ change, there will be a flow of capital and labor between enterprise 1 and enterprise 2. We measure the changes in capital and labor through $x_{k,t}$ and $x_{l,t}$. The process of reallocating production factors also reflects the process of industrial structural transformation and upgrading.

Combined with Eq (30) with (10), (11) and (17), the functional form for the capital and labor proportions of enterprise 1 is as follows:

$$x_{k,t} = \frac{(N_t - I_t)\gamma Y_{1,t}^{\frac{\varepsilon-1}{\varepsilon}}}{(N_t - I_t)\gamma Y_{1,t}^{\frac{\varepsilon-1}{\varepsilon}} + (1-\alpha)(1-\gamma)Y_{2,t}^{\frac{\varepsilon-1}{\varepsilon}}}, \tag{32}$$

$$x_{l,t} = \frac{(N_t - I_t)\gamma Y_{1,t}^{\frac{\varepsilon-1}{\varepsilon}}}{(N_t - I_t)\gamma Y_{1,t}^{\frac{\varepsilon-1}{\varepsilon}} + (1-\alpha)(1-\gamma)Y_{2,t}^{\frac{\varepsilon-1}{\varepsilon}}}. \tag{33}$$

To facilitate calculation, the ratio of the relative amount of labor and capital is as follows.

$$l_t \equiv \frac{L_{1,t}}{L_{2,t}} = \frac{N_t - I_t}{1 - \alpha} \cdot \frac{\gamma}{1 - \gamma} \left(\frac{Y_{1,t}}{Y_{2,t}}\right)^{\frac{\varepsilon-1}{\varepsilon}}, \tag{34}$$

$$k_t \equiv \frac{K_{1,t}}{K_{2,t}} = \frac{I_t - N_t + 1}{\alpha} \cdot \frac{\gamma}{1 - \gamma} \left(\frac{Y_{1,t}}{Y_{2,t}}\right)^{\frac{\varepsilon-1}{\varepsilon}}. \tag{35}$$

It can be seen from (32)–(35) that $x_{k,t} = \frac{k_t}{k_t+1}$, $x_{l,t} = \frac{l_t}{l_t+1}$. $x_{k,t}$ and $k_t$ have the same monotonicity, and $x_{l,t}$ and $l_t$ also have the same monotonicity.

**Proposition 2.**

(i) When $\varepsilon > 1$, the AI enterprise and the non-AI enterprise are in the same industry. As the degree of automation deepens ($I_t$ becomes larger), the change of $x_{l,t}$ and $l_t$ is uncertain, and the relative amount of capital $k_t$ increase. As the number of new tasks increases ($N_t$ becomes larger), $x_{l,t}$ and $l_t$ increase, and the change in the relative amount of capital $k_t$ is uncertain. When the changes in the automation deepening and the new tasks are the same ($dI_t = dN_t$), the relative amount of labor $l_t$ and the relative amount of capital $k_t$ increase.

(ii) When $0 < \varepsilon < 1$, the AI enterprise and the non-AI enterprise are in different industries. As the degree of automation deepens ($I_t$ becomes larger), $x_{l,t}$ and $l_t$ decrease, while the change in the relative amount of capital $k_t$ is uncertain. As the number of new tasks increases ($N_t$ becomes larger), the change of $x_{l,t}$ and $l_t$ is uncertain, while the relative amount of capital $k_t$ decrease. When the changes in the automation deepening and the new task are the same ($dI_t = dN_t$), the relative amount of labor $l_t$ and the relative amount $k_t$ of the increase.

(All proofs are contained in the S1 Appendix.).

The conclusion of Proposition 2 is consistent with that of Proposition 1, while the corresponding flowchart is shown in Figs 1 and 2. In the same industry, the AI enterprise attracts more labor and capital into the AI enterprise due to improvement of production efficiency and R&D efficiency. The non-AI enterprise loses its advantages in the competition and gradually becomes eliminated. The use of artificial intelligence has improved the technical level of the industry and promoted the transformation and upgrading of the industrial structure of the same industry. In different industries, due to the impact of the consumption effect, capital and labor flow from the AI enterprise to the non-AI enterprise. On the contrary, the industry that

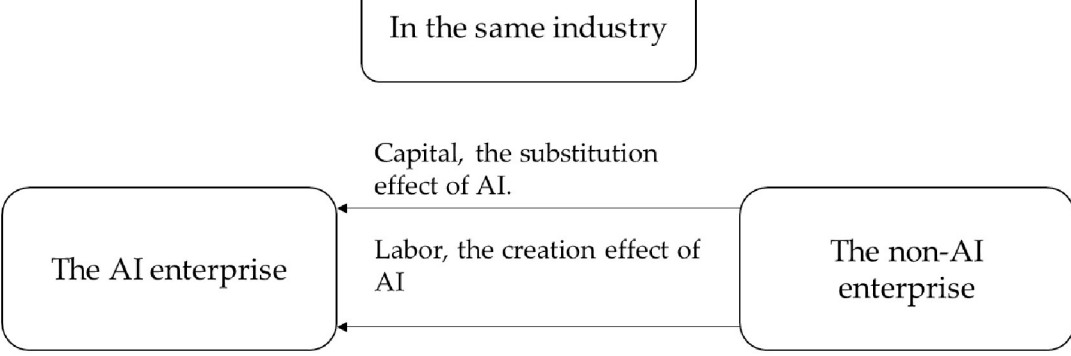

**Fig 1. In the same industry, capital flows between the AI enterprise and the non-AI enterprise.**

**Fig 2. In different industries, capital flows between the AI enterprise and the non-AI enterprise.**

uses artificial intelligence is restrained, which will affect the improvement of the production level of the entire society.

## Dynamic equilibrium

Suppose that the population growth rate is $n$, and the population at time 0 is $L(0)$, the population size at time $t$ is

$$L(t) = L(0)e^{nt}. \tag{36}$$

The utility function of a representative family is

$$U = \int_0^\infty \left(\frac{C^{1-\theta}-1}{1-\theta}\right)e^{-\rho t}dt, \tag{37}$$

where $\theta$ is the relative risk aversion factor and $\rho$ is the subjective time discount rate. The family budget constraint is the following.

$$\dot{K} = rK + WL - C. \tag{38}$$

In order to prevent the *Ponzi* scheme, the lifetime consumption cannot be greater than the income of a lifetime, so the cross-sectional condition is as follows.

$$\lim_{t\to\infty}\{K(t) \cdot exp[-\int_0^t (r(v) - n)dv]\} \geq 0. \tag{39}$$

The Euler equation is

$$\frac{\dot{C}}{C} = \frac{r - \rho}{\theta}, \tag{40}$$

For simplicity, note the growth rate as follows: $\frac{\dot{L_1}}{L_1} \equiv n_1, \frac{\dot{L_2}}{L_2} \equiv n_2, \frac{\dot{L}}{L} \equiv n, \frac{\dot{K_1}}{K_1} \equiv m_1, \frac{\dot{K_2}}{K_2} \equiv m_2, \frac{\dot{K}}{K} \equiv m, \frac{\dot{I}}{I} \equiv g_I, \frac{\dot{N}}{N} \equiv g_N, \frac{\dot{A_2}}{A_2} = \sigma, \frac{\dot{Y_1}}{Y_1} \equiv g_1, \frac{\dot{Y_2}}{Y_2} \equiv g_2, \frac{\dot{Y}}{Y} \equiv g$.

When time is infinite, the above growth rates are called the progressive growth rate, which can be recorded as $n_s^* = \lim_{t\to\infty} n_s, z_s^* = \lim_{t\to\infty} z_s, g_s^* = \lim_{t\to\infty} g_s, s = 1, 2 \ldots$

**Proposition 3.** The progressive growth rates $g_1^*$, $g_2^*$ exit, if $\varepsilon > 1$, then $g^* = max\{g_1^*, g_2^*\}$. If $0 < \varepsilon < 1$, then $g^* = min\{g_1^*, g_2^*\}$. When $dN = dI$, we have

$$g_1^* = ln\frac{N}{N-1} + (I - N + 1)m_1 + (N - I)n_1, \tag{41}$$

$$g_2^* = \sigma + \alpha m_2 + (1 - \alpha)n_2. \tag{42}$$

(All proofs are contained in the S1 Appendix.).

When $\varepsilon > 1$, the AI enterprise and the non-AI enterprise are in the same industry, the enterprises that adopt automation strategies play an important role and determine the rate of economic growth of the industry. When $0 < \varepsilon < 1$, the AI enterprise and the non-AI enterprise are in different industries, the economic growth rates are decided by the non-AI enterprise, which developed slower. As we showed above, enterprises in the same industry compete with each other. When an enterprise adopts the automation strategy, others follow. When enterprises are in different industries, enterprises that do not adopt an automation strategy gain more technical dividends due to technological externalities. Thereafter, capital and labor flow into the non-AI enterprise, which is generally a labor-intensive industry. The non-AI enterprise has no incentive to innovate since it can freely ride the benefits. In the long run, externality can become a factor that hinders social development.

## National macrocontrol policy

According to the above analysis, the overall social economic growth in different industries depends on the slower development of the non-AI enterprise. This will become resistance to social progress and economic development. The government must adopt macroeconomic policies to reduce externalities and encourage enterprises to carry out technological innovation.

In reality, the government levies taxes on both the AI and non-AI enterprises. However, when the government adopts subsidy policies, it essentially equates to providing tax relief to the AI enterprise. This subsidy policy fundamentally involves utilizing tax revenue collected from non-AI enterprise to support the AI enterprise. To simplify the analysis, this paper assumes that the government imposes taxes on the non-AI enterprise and subsequently utilizes these tax proceeds to subsidize the AI enterprise. Another rationale for employing the aforementioned analysis is that artificial intelligence exhibits spillover effects. This implies that the non-AI enterprise benefits from positive externalities arising from the innovation and development of the AI enterprise. Hence, the government can rectify this situation by imposing corrective taxes.

The following only consider different industries, so $0 < \varepsilon < 1$. The government levies a correction tax $\tau_2$ on the output of the non-AI enterprise and subsidizes the AI enterprise to purchase automation equipment. The subsidy rate is $\tau_1$. Government fiscal balance, thus

$$\tau_1 R_{1,t} K_{1,t} = \tau_2 p_{2,t} Y_{2,t}, \tag{43}$$

The AI enterprise only considers the impact of subsidies on the automation process from its own perspective. If there is no subsidy, at the critical point $I$, AI enterprises use capital and labor at the same cost. ie $\frac{R_{1,t}}{\gamma_{k_1}(I_t)} = \frac{W_{1,t}}{\gamma_{l_1}(I_t)}$, and the deformation is $\frac{\gamma_{l_1}(I_t)}{\gamma_{k_1}(I_t)} = \frac{W_{1,t}}{R_{1,t}}$. After the government subsidizes, the equilibrium point changes to $I_{\tau_1}$, that is, $\frac{(1-\tau_1)R_{1,t}}{\gamma_{k_1}(I_{\tau_1})} = \frac{W_{1,t}}{\gamma_{l_1}(I_{\tau_1})}$, and the deformation is $\frac{\gamma_{l_1}(I_{\tau_1})}{\gamma_{k_1}(I_{\tau_1})} = \frac{W_{1,t}}{(1-\tau_1)R_{1,t}}$, so $\frac{\gamma_{l_1}(I_t)}{\gamma_{k_1}(I_t)} < \frac{\gamma_{l_1}(I_{\tau_1})}{\gamma_{k_1}(I_{\tau_1})}$. Since $\gamma_{l_1}(i)/\gamma_{k_1}(i)$ is monotonically increasing on $I$, so $I_t < I_{\tau_1}$, that is, the government's subsidy policy for the AI enterprise will indeed encourage enterprises to buy automation equipment and accelerate machine substitution. Of course, this

is only from the production point of view. The above promotion of enterprises to improve the level of AI application through subsidies is called the productivity effect.

If both industries are considered at the same time, richer results will be produced. In this section, to examine the impact of subsidy policies on employment, we define the unemployment rate as $e_t$. Given that artificial intelligence is expected to replace a significant portion of the workforce, we assume that the unemployed individuals originate from the pool of labor employed by intelligent enterprises. Initially, the labor force of intelligent enterprise 1 is denoted as $L_{1,t}$, and the number of unemployed individuals is $e_t L_{1,t}$. As a result, the workforce of enterprise 1 subsequently becomes $(1 - e_t)L_{1,t}$. Based on Frey & Osborne [48], the probability of worker unemployment risk is denoted as $r_{s,t} = \frac{1}{1+\exp(-I_t)}$. The probability of unemployment risk, as expressed in the aforementioned form, ensures both $0 < r_{s,t} < 1$ and a positive correlation between the unemployment risk probability $r_{s,t}$ and the level of automation $I_t$. It should be explicitly noted that the probability of unemployment risk $r_{s,t}$ pertains to a priori probability, whereas the unemployment rate $e_t$ pertains to a posteriori probability, and these two are not the same.

The AI enterprise receives subsidies and pursues profit maximization.

$$\max_{K_{1,t},L_{1,t}} \pi_{1,t} = p_{1,t}Y_{1,t} - W_{1,t}(1 - e_t)L_{1,t} - (1 - \tau_1)R_{1,t}K_{1,t}, \tag{44}$$

Obtained by first-order conditions.

$$W_{1,t} = (N_t - I_t)p_{1,t}Y_{1,t}/((1 - e_t)L_{1,t}), R_{1,t} = (I_t - N_t + 1)p_{1,t}Y_{1,t}/[(1 - \tau_1)K_{1,t}]. \tag{45}$$

The non-AI enterprise is taxed,

$$\max_{K_{2,t},L_{2,t}} \pi_{2,t} = (1 - \tau_2)p_{2,t}Y_{2,t} - W_{2,t}L_{2,t} - R_{2,t}K_{2,t}, \tag{46}$$

$$W_{2,t} = (1 - \tau_2)(1 - \alpha)p_{2,t}Y_{2,t}/L_{2,t}, R_{2,t} = (1 - \tau_2)\alpha p_{2,t}Y_{2,t}/K_{2,t}. \tag{47}$$

In the equilibrium state, the wages of the two enterprises will be the same and the gains of capital in the two industries will be the same.

$$W_{1,t} = W_{2,t}, R_{1,t} = R_{2,t}, \tag{48}$$

Obtained by (45), (47) and (48), then

$$y_t^{\frac{\varepsilon-1}{\varepsilon}} = \frac{1-\gamma}{\gamma} \cdot \frac{(1-\tau_2)(1-\alpha)}{N_t - I_t}(1 - e_t)l_t, \tag{49}$$

$$(1 - \tau_1)k_t = \frac{1-\alpha}{\alpha} \cdot \frac{I_t - N_t + 1}{N_t - I_t}(1 - e_t)l_t. \tag{50}$$

where $l_t \equiv \frac{L_{1,t}}{L_{2,t}}, k_t \equiv \frac{K_{1,t}}{K_{2,t}}, y_t \equiv \frac{Y_{1,t}}{Y_{2,t}}$.

**Proposition 4.** The government imposes a tax rate of $\tau_2$ on the output of the non-AI enterprise and subsidizes the purchase of automation equipment by the AI enterprise. As the tax rate $\tau_2$ increases, the relative quantity $k$ of capital will increase, the unemployment rate will increase, and the relative quantity of labor $l_t$ will decrease. Task $N_t$ will increase, while artificial intelligence level $I_t$ will decrease; relative output quantity $y_t$ will increase.

(All proofs are contained in the S1 Appendix.).

In various industry contexts, when non-AI enterprises are taxed while AI enterprises receive subsidies, Proposition 4 illustrates the mobility of capital and labor in Fig 3, leading to the following economic impacts:

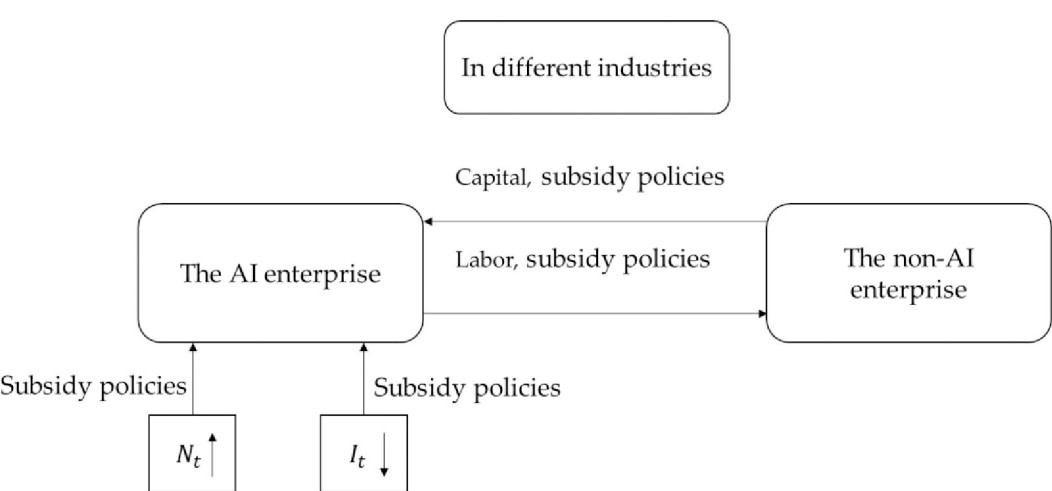

**Fig 3. The impact of subsidy policies on capital and labor mobility.**

(i) Funds shift from the non-AI enterprise to the AI enterprise, while labor transitions from the AI enterprise to the non-AI enterprise. The subsidy policy lowers the cost of AI equipment for AI enterprises, initially encouraging greater adoption of AI technology and expediting the process of AI integration. As a consequence, surplus labor from the AI enterprise finds employment in the non-AI enterprise. In the initial stage, the AI enterprise accumulates more capital, while the non-AI enterprise acquires a larger labor force.

(ii) The subsidy policy accelerates the AI adoption process within enterprises, leading to improved production efficiency and output. The share of output produced by the AI enterprise increases.

(iii) After a certain period, the pace of AI adoption starts to decelerate. The AI enterprise must generate new tasks to meet the employment needs of the workforce. Given that products from both sectors are complementary and challenging to replace—such as automobiles and food—even though the AI enterprise enhances its production, the non-AI enterprise experiences reduced capital due to taxation, limiting its production capacity. The interdependence of these two product types implies that society cannot indefinitely consume the products of the AI enterprise. Over time, due to overproduction by the AI enterprise, the pace of AI adoption slows down. Simultaneously, with decreased capital, the non-AI enterprise faces a relative surplus of labor supply, compelling the AI enterprise to create more job opportunities to meet the workforce's employment requirements.

The positive effects of subsidy policies include:

(i) Stimulating intelligent enterprises to procure more intelligent equipment, thereby accelerating the process of automation and enhancing production efficiency. This aids in improving product quality and reducing production costs, subsequently increasing the competitiveness of enterprises. (ii) Significantly boosting the output of intelligent enterprises, allowing them to offer more high-quality products and services in both domestic and international markets. This is likely to increase export volumes, promote GDP growth, and inject vitality

into economic development. (iii) Subsidy policies encourage non-intelligent enterprises to actively adopt intelligent strategies, raising the overall level of automation in society. This, in turn, promotes industrial upgrading, creates more high-skilled jobs, offers greater development opportunities for workers, and contributes to economic growth and sustainability across society.

Negative impacts of subsidy policies include:

(i)Some non-intelligent enterprises may find it challenging to transition to automation, particularly in sectors such as communication and courier services. These enterprises may face competitive pressures, leading to business contraction or closure, which could affect employment and local economies. (ii)Subsidy policies do not guarantee a continuous and uninterrupted pace of automation. Over time, the rate of automation adoption may gradually decelerate. This may result in reduced investment demands for intelligent enterprises and impact the development of related industrial supply chains. (iii)In the long term, as the level of automation increases, there may be a significant risk of widespread unemployment. Some workers may need to acquire new skills to adapt to evolving job requirements, which can be a time-consuming and resource-intensive process. Governments will need to formulate appropriate policies to assist these workers in retraining and securing reemployment opportunities.

## Future direction

With the development of technologies such as artificial intelligence and big data, the future development direction of society is that there are machine substitutions in many fields. In real life, not only is the manufacturing industry gradually automating, but there are also signs of machine substitution in the service industry.

Next, this article examines the use of artificial intelligence for production in both industries. Due to the gradual displacement of non-AI enterprise by AI enterprise in the same industry, this section only discusses situations in different industries, i.e., where $\varepsilon < 1$. The AI Enterprise 1 is the same as (5), (6), (7) and (8) above. The non-AI enterprise 2 adopts automated production, Enterprise 2's production task is $i \in [M_t - 1, M_t]$. For tasks $i \in [M_t - 1, J_t]$, where artificial intelligence has been implemented and can be produced by either workers or machines, and for tasks $i \in (J_t, M_t]$, where AI implementation has not occurred, and production depends solely on workers, the production function for enterprise 2 can be obtained using the same analytical methods as enterprise 1

$$Y_{2,t} = A_{2,t} K_{2,t}^{J_t - M_t + 1} L_{2,t}^{M_t - J_t}, \tag{51}$$

Where $A_{2,t} = \frac{\exp(\int_{M_t-1}^{J_t} \ln \gamma_{k_2}(i)di + \int_{J_t}^{M_t} \ln \gamma_{l_2}(i)di)}{(J_t - M_t + 1)^{J_t - M_t + 1}(M_t - J_t)^{M_t - J_t}}$ is the overall technical level of Enterprise 2. Where $\gamma_{l_2}(i)$ is the production efficiency of the labor force in task $i$, and $\gamma_{k_2}(i)$ is the production efficiency of the intelligent machine in task $i$. The capital of Enterprise 2 is $K_{2,t} = \int_{M_t-1}^{J_t} k_{2,t}(i)di$, and the labor force is $L_{2,t} = \int_{J_t}^{M_t} l_{2,t}(i)di$. The capital share is $J_t - M_t + 1$ and the labor share is $M_t - J_t$. Because Enterprise 2 adopts the machine substitution strategy, the capital share increases, the labor income share decreases, and the labor income share will be less than the capital share, so it is assumed that $0 < M_t - J_t < 1/2$. Because Enterprise 2 adopted artificial intelligence later than Enterprise 1, Enterprise 1 has a higher capital share than Enterprise 2, thus $M_t - J_t > N_t - I_t$.

If the wage for workers at enterprise 2 is denoted as $W_{2,t}$, and the cost (or rental price) of intelligent equipment is represented as $R_{2,t}$, then:

$$W_{2,t} = (M_t - J_t)p_{2,t}Y_{2,t}/L_{2,t}, \tag{52}$$

$$R_{2,t} = (J_t - M_t + 1)p_{2,t}Y_{2,t}/K_{2,t}, \tag{53}$$

In the equilibrium state, the wages of the two enterprises will be the same and the gains of capital in the two industries will be the same.

$$W_{1,t} = W_{2,t}, \; R_{1,t} = R_{2,t}, \tag{54}$$

From (10), (11), (52), (53) and (54)

$$l_t = \frac{N_t - I_t}{I_t - N_t + 1} \cdot \frac{J_t - M_t + 1}{M_t - J_t} k_t, \tag{55}$$

$$\frac{x_{l,t}}{1 - x_{l,t}} = \frac{N_t - I_t}{I_t - N_t + 1} \cdot \frac{J_t - M_t + 1}{M_t - J_t} \frac{x_{k,t}}{1 - x_{k,t}}. \tag{56}$$

Where $l_t \equiv \frac{L_{1,t}}{L_{2,t}}, k_t \equiv \frac{K_{1,t}}{K_{2,t}}, x_{k,t} = \frac{K_{1,t}}{K_t}, \; x_{l,t} = \frac{L_{1,t}}{L_t}$.

Similar to the analysis above

$$V_{1,t} = \frac{\pi_{k_{1,t}}}{r(t)} = (N_t - I_t)K_{1,t}, \; V_{2,t} = \frac{\pi_{k_{2,t}}}{r(t)} = (M_t - J_t)K_{2,t}, \tag{57}$$

$$v_t \equiv \frac{V_{1,t}}{V_{2,t}} = \frac{N_t - I_t}{M_t - J_t} k_t. \tag{58}$$

From (56),

$$\frac{dln x_{k,t}}{dln(A_{1,t}/A_{2,t})} = \frac{(1 - x_{k,t})(\varepsilon - 1)}{1 + (\varepsilon - 1)(M_t - J_t - N_t + I_t)(x_{k,t} - x_{l,t})} < 0 \tag{59}$$

$$\frac{dln x_{l,t}}{dln(A_{1,t}/A_{2,t})} = \frac{(1 - x_{l,t})(\varepsilon - 1)}{1 + (\varepsilon - 1)(M_t - J_t - N_t + I_t)(x_{k,t} - x_{l,t})} < 0 \tag{60}$$

$$\frac{dln x_{k,t}}{dln(K)} = \frac{(1 - x_{k,t})(M_t - J_t - N_t + I_t)(\varepsilon - 1)}{1 + (\varepsilon - 1)(M_t - J_t - N_t + I_t)(x_{k,t} - x_{l,t})} < 0 \tag{61}$$

$$\frac{dln x_{l,t}}{dln(K)} = \frac{(1 - x_{l,t})(M_t - J_t - N_t + I_t)(\varepsilon - 1)}{1 + (\varepsilon - 1)(M_t - J_t - N_t + I_t)(x_{k,t} - x_{l,t})} < 0 \tag{62}$$

**Proposition 5.**

(i) When Enterprise 1 enhances its artificial intelligence technology, capital and labor flow from Enterprise 1 to Enterprise 2. When Enterprise 2 improves its artificial intelligence technology, capital and labor flow from Enterprise 2 to Enterprise 1.

(ii) When the total capital stock increases, capital and labor shift from Enterprise 1 to Enterprise 2.

Changes in technological measures will lead to adjustments in an enterprise's artificial intelligence technologies, thereby resulting in varying levels of development across different

departments. Such technological shifts will impact the enterprise's output, ultimately causing fluctuations in product prices and subsequently prompting the flow of capital and labor. Using the manufacturing and service industries as examples, when the manufacturing industry enhances its production efficiency through the utilization of artificial intelligence technologies, it experiences an increase in output, subsequently leading to lower product prices. However, due to relatively stable price demand, individuals do not purchase more manufacturing products simply because they have become more affordable. This results in a reduction in the prices of manufacturing products and a relative increase in the prices of products in the service industry. The price effect then steers capital and labor from the manufacturing industry to the service industry.

Conversely, when the service industry improves its efficiency through the adoption of artificial intelligence technologies, capital and labor flow from the service industry to the manufacturing industry. An increase in the overall capital stock prompts capital and labor to migrate from industries with higher technological sophistication to those with lower technological advancement.

This transformation underscores the fact that societal development depends not only on its areas of expertise but also on its areas of relative deficiency. When society's production factors are relatively abundant, the price effect guides production factors from industries with higher technological proficiency to those with lower technological sophistication, thus promoting balanced development across the entire society.

This article defines technology measurement to measure economic and social development. Technology measurement should not only reflect production efficiency (reflected by $I_t$ and $J_t$), but also include labor share (reflected by $N_t$ and $M_t$). Therefore, the technical measures of the two enterprises in this paper are $\chi_{1,t} = N_t - I_t, \chi_{2,t} = M_t - J_t, 0 < \chi_{1,t}, \chi_{2,t} < 1/2$. The introduction of new AI technologies (when $I_t$ and $J_t$ increase) will reduce the technical measures, and the introduction of new tasks (when $N_t$ and $M_t$ increase) will increase the technical measures. The development of both technologies will improve production efficiency and increase total social output. The deepening of AI technology ($I_t$ and $J_t$ become larger) will replace human jobs and reduce the share of labor income. The increase of new tasks ($N_t$ and $M_t$ become larger) will increase the number of human jobs and increase the share of labor income.

In order to simplify the analysis, this article assumes that artificial intelligence technology has the same productivity on different tasks, that is, $\gamma_{l_1} = \gamma_{l_1}(i), \gamma_{k_1} = \gamma_{k_1}(i), \gamma_{l_2} = \gamma_{l_2}(i), \gamma_{k_2} = \gamma_{k_2}(i)$. When the technical measure increases, the enterprise's labor share will increase, that is, the increase in technical measure tends to increase labor productivity. In order to reflect the above characteristics, this article assumes

$$\frac{\gamma_{l_1}}{\gamma_{k_1}} > \frac{x_{1,t}}{1 - x_{1,t}}, \frac{\gamma_{l_2}}{\gamma_{k_2}} > \frac{x_{2,t}}{1 - x_{2,t}} \tag{63}$$

The expressions for the total labor share and capital share are as follows:

$$\alpha_{L,t} = \frac{\omega_{1,t} L_{1,t} + \omega_{2,t} L_{2,t}}{Y_t} = \frac{x_{1,t}\left(\frac{Y_{1,t}}{Y_{2,t}}\right)^{\frac{\varepsilon-1}{\varepsilon}} + x_{2,t}}{\left(\frac{Y_{1,t}}{Y_{2,t}}\right)^{\frac{\varepsilon-1}{\varepsilon}} + \frac{1-\gamma}{\gamma}} \tag{64}$$

$$\alpha_{K,t} = \frac{R_{1,t} K_{1,t} + R_{2,t} K_{2,t}}{Y_t} = 1 - \frac{x_{1,t}\left(\frac{Y_{1,t}}{Y_{2,t}}\right)^{\frac{\varepsilon-1}{\varepsilon}} + x_{2,t}}{\left(\frac{Y_{1,t}}{Y_{2,t}}\right)^{\frac{\varepsilon-1}{\varepsilon}} + \frac{1-\gamma}{\gamma}} \tag{65}$$

**Proposition 6.**

(i) When the technological measure $\chi_{1,t}$ increases, the research and development value $V_{1,t}$ of firm 1 increases at a faster rate, leading to an increase in the labor proportion and a decrease in the capital proportion of firm 1. Conversely, when the technological measure $\chi_{2,t}$ increases, the research and development value $V_{2,t}$ of firm 2 increases at a faster rate, resulting in a decrease in the labor proportion and an increase in the capital proportion of firm 1.

(ii) When $d\chi_{1,t} = d\chi_{2,t}$, if $\chi_{2,t} > \chi_{1,t}$, then $dv_t > 0$, $dx_{l,t} > 0$, $dx_{k,t} < 0$. If $\chi_{2,t} = \chi_{1,t}$, then $dv_t = dx_{l,t} = dx_{k,t} = 0$. If $\chi_{2,t} < \chi_{1,t}$, then $dv_t < 0$, $dx_{l,t} < 0$, $dx_{k,t} > 0$.

(iii) When enterprises adopt artificial intelligence technology that is biased towards improving labor productivity, increasing technical measurement will improve the level of artificial intelligence technology, thereby increasing enterprise output. When enterprises increase their technical measures, the total labor share will increase and the total capital share will decrease.

(All proofs are contained in the S1 Appendix.).

In Figs 4 and 5, when technological measure $\chi_{1,t}$ increases, capital flows from an AI enterprise to a non-AI enterprise, while labor moves from a non-AI enterprise to an AI enterprise. When technological measure $\chi_{2,t}$ increases, capital and labor experience the opposite flow.

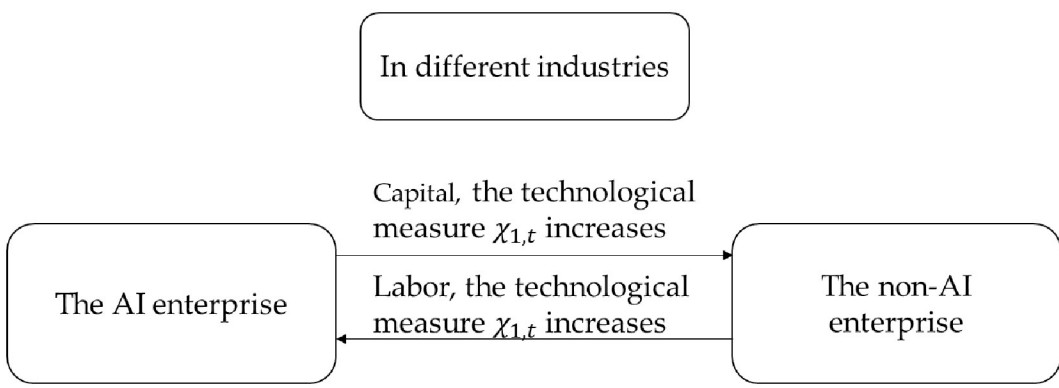

**Fig 4. When technological measures $\chi_{1,t}$ increase, capital and labor mobility between AI and non-AI enterprises.**

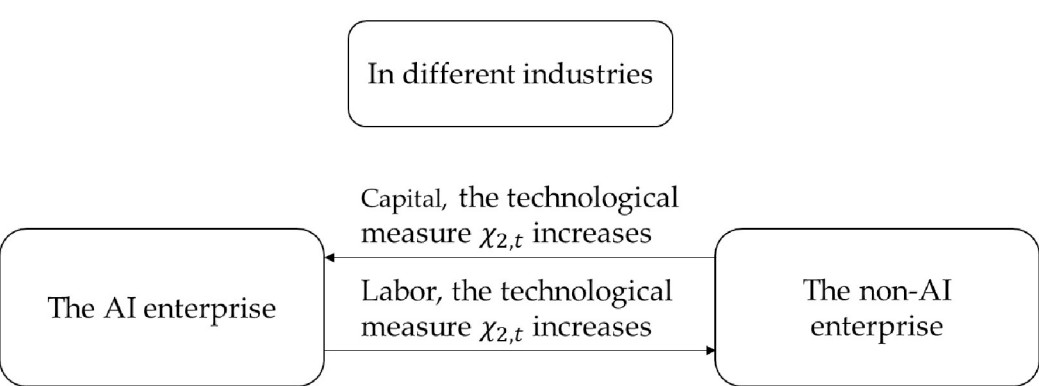

**Fig 5. When technological measures $\chi_{2,t}$ increase, capital and labor mobility between AI and non-AI enterprises.**

The economic intuitions underlying Proposition 6 can be summarized as follows:

1. An increase in technological measures leads to an increase in the R&D value for firms because it increases the number of labor tasks. This implies a positive relationship between technological measures and R&D value. When technological measures improve, firms will rely more on labor rather than capital. For example, an increase in technological measures for Firm 1 will increase the demand for labor, attracting labor from Firm 2. Meanwhile, Firm 1's demand for capital decreases, leading to a flow of capital toward Firm 2.

2. When both firms experience the same level of technological advancement, the initial size of their technological measures determines the flow of labor and capital. If Firm 2's technological measure is greater than that of Firm 1, then the R&D value of Firm 1 will be greater than that of Firm 2. Since R&D value tends to favor labor, Firm 1 will have a greater demand for labor, attracting labor from Firm 2. At the same time, Firm 1's demand for capital decreases, causing capital to flow from Firm 1 to Firm 2.

3. When both firms have the same technological measures, the demand for labor and capital is equal for both Firm 1 and Firm 2, leading to a halt in the flow of labor and capital. This suggests that the development of artificial intelligence technology tends to lead to convergence, where firms with different initial technological measures gradually approach equal levels of labor and capital shares.

These intuitive points emphasize the impact of technological measures on R&D value, labor demand, and capital demand for firms, as well as the dynamics of labor and capital flow between firms with different technological measures. These insights contribute to a better understanding of the relationship between technological measures and economic development.

Technological convergence offers several advantages, including the enhancement of labor-biased technologies, the promotion of labor income share stability, and the reduction of income inequality. Nevertheless, it may simultaneously exert adverse influences on innovation and competition.

Primarily, technological convergence implies a relatively narrow technological gap among enterprises, fostering a balanced labor market. Under such circumstances, scenarios where certain enterprises exhibit an exceptionally high demand for labor, leading to wage inflation, while others demonstrate comparatively lower labor demands, resulting in wage disparities, are less likely to occur. This equilibrium in the labor market contributes to ensuring a more widespread distribution of the benefits of economic growth, consequently stabilizing labor income shares. Furthermore, the diminished prominence of technology as a distinguishing factor enables a broader pool of talent and labor to access employment opportunities across diverse enterprises, thus mitigating wage disparities and socio-economic inequality. This, in turn, promotes social stability and sustainable development.

However, technological convergence may introduce certain unfavorable consequences. Primarily, it could diminish the incentive for enterprises to engage in innovative activities. With the technological levels among enterprises approaching parity, businesses may be more inclined to maintain their existing technological status, rather than committing resources to risky innovative research and development. This could lead to a decrease in innovation, as the pursuit of innovation typically demands higher risks and investment, which may no longer appear as financially advantageous in markets with similar technological standings.

Moreover, the convergence of technology implies that enterprises are no longer able to maintain a distinctive technological advantage, resulting in heightened competitive pressures. Faced with this reality, businesses must rely on alternative differentiation strategies such as

pricing, marketing approaches, and service quality to set themselves apart. The intensified competitive environment may encourage enterprises to implement cost-cutting measures, which could encompass reductions in workforce or wage levels to sustain competitiveness, thereby potentially affecting labor stability adversely.

In summary, technological convergence offers a measure of labor market stability and inequality reduction. However, it may concurrently pose negative implications for innovation and competition. When formulating policies and managing enterprises, it is imperative to diligently weigh these factors to achieve sustainable economic and social growth in accordance with academic norms and standards.

### Ethics statement

This article utilizes data from the China Statistical Yearbook, and we solemnly pledge our adherence to ethical principles and legal regulations, including the lawful acquisition of data, privacy protection, academic integrity, source citation, and research transparency. We commit to handling the data with honesty and care, respecting intellectual property rights and data providers' interests, thereby ensuring the integrity and credibility of the research.

### Data extraction

Data for this study was extracted from the China Statistical Yearbook, a reputable and publicly available source widely recognized for its comprehensive and reliable statistical data pertaining to various aspects of China's economy, society, and demographics. The selected datasets were collected for the specific purpose of this research, and meticulous care was taken to ensure accuracy and relevance to the study's objectives. Our minimal dataset can be downloaded at https://figshare.com/account/home. The data is published in Figshare with the file name The minimum data set.zip.

## Results

The main conclusions of this paper are Propositions 1 to 6. In this section, we will employ numerical simulation methods to validate the accuracy of Propositions 1 to 6.

### Numerical simulation

The emergence of artificial intelligence in China commenced around 2010. Consequently, the simulations in this article commence from the year 2011 and extend for 30 periods, with each period representing one year. To reflect the equal attention to the two industries, $\gamma = 0.5$. According to Lankisch et al. (2019), the capital depreciation rate $\delta$ is 0.05 and the savings rate $s$ is 0.21. The labor share of the AI enterprise 1 is $N_t - I_t$, where $I_t$ reflects the process of intelligence, while $N_t$ reflects the generation of new tasks. This paper will discuss in two cases: (1) $N_t$ remains constant, while $I_t$ increases. (2) $I_t$ remains constant, while $N_t$ increases.

This paper primarily investigates the development status of two sectors: manufacturing and services, as they currently contribute significantly to China's GDP. According to data from the National Bureau of Statistics of China in 2011, the value-added by the secondary sector accounted for 46.5% of GDP, the tertiary sector accounted for 44.2%, while the primary sector accounted for only 9.3%. Furthermore, over time, the contribution of the primary sector has been decreasing annually and in 2021, it accounted for just 7.3%. According to data from the National Bureau of Statistics in 2010, the population engaged in manufacturing was 218 million, while the population engaged in the service sector was 263 million. This paper primarily examines the impact of artificial intelligence on the economy, disregarding demographic

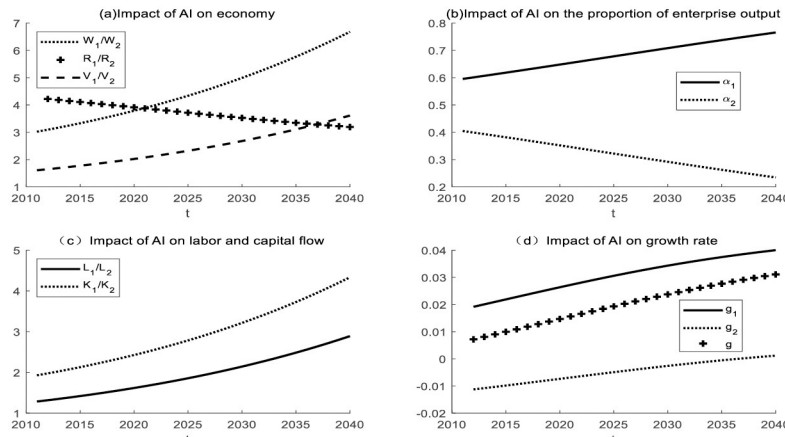

**Fig 6. The impact of AI on two enterprises in the same industry with the same change range of $N_t$ and $I_t$.**

factors. Therefore, we assume a population growth rate of 0. In sensitivity analysis, a population growth rate of -0.3% is set, but its impact on the results can be considered negligible.

**The change range of $N_t$ and $I_t$ is the same, and the labor share $N_t - I_t$ of AI enterprise remains unchanged.** (1)Impact of AI on enterprises in the same industry

In the same industry, the products of Enterprise 1 and Enterprise 2 are interchangeable, so $\varepsilon > 1$. Enterprise 1 uses artificial intelligence for production, and enterprise 2 uses traditional material equipment for production. Take $\varepsilon = 5$ for the simulation. The initial assumptions are made that the labor force of the two enterprises is heterogeneous and that the production equipment used by the two enterprises is different. The simulation results are shown in Fig 6.

It can be seen in Fig 6(a) and 6(b) that in the same industry, the AI enterprise will have a significant advantage. Compared to non-AI enterprises, AI enterprise1 grows faster in output, wages, R&D value, and capital stock. The proportion of the output of the AI enterprise increased, while that of non-AI enterprise decreased. It can also be seen from Fig 1 that the use of AI reduces the price of intelligent devices, which is in line with Moore's law. That is, the price of high-tech products will drop by about half every other period of time. AI reduces the price of intelligent equipment and is conducive to the large-scale use of intelligent equipment by enterprises. This will promote the survival of the fittest in the industry and facilitate industrial transformation and upgrading.

If the labor force of two enterprises is assumed to be homogeneous and capital has the same rate of return, the flow of labor force and capital can be observed through numerical simulation. The simulation results are shown in Fig 6(c) and 6(d). Labor and capital in the same industry will flow from non-AI enterprise 2 to AI enterprise 1. At the same time, enterprises that use smart devices will achieve a higher growth rate. As time goes by, economic growth will tend to be stable due to the diminishing scale effect. In reality, enterprises that take the lead in using intelligent devices in the manufacturing industry will become leaders in the industry. The AI enterprise will attract more capital and labor to join, promote the survival of the fittest in the industry, and facilitate industrial transformation and upgrading.

(2) Impact of AI on enterprises in different industries

In different industries, the products of Enterprise 1 and Enterprise 2 are difficult to replace each other, so $0 < \varepsilon < 1$. For example, Enterprise 1 is a manufacturing industry and Enterprise 2 is a service industry. Take $\varepsilon = 0.5$ for simulation. It is initially assumed that the labor force of the two enterprises is heterogeneous. The simulation results are shown in Fig 7. In different

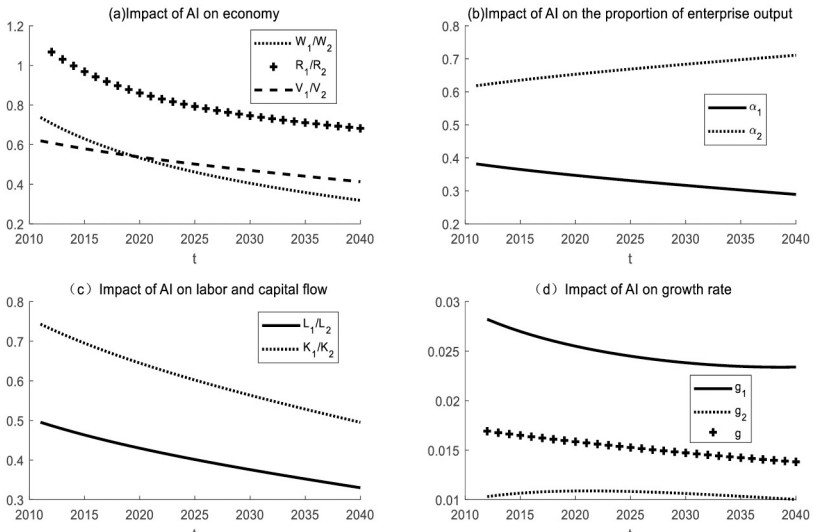

**Fig 7. Impact of AI on different industries with the same change range of $N_t$ and $I_t$.**

industries, non-AI enterprises 2 benefit more. The initial stage will fluctuate under the influence of the initial value setting. After a period of time, the ratio of production, wage, and relative value of the R&D value will decline. The proportion of output of AI enterprises decreased, while that of non-AI enterprises increased.

If the labor force of two enterprises is assumed to be homogeneous and capital has the same rate of return, the flow of labor force and capital can be observed through numerical simulation. The simulation results are shown in Fig 7(c) and 7(d). In different industries, labor and capital will flow from AI enterprise to non-AI enterprise. Enterprise 2 has low production efficiency, but attracts capital and labor inflow due to consume effect. This will lead to a decline in the production efficiency of enterprise 1, which is not conducive to the development of a social economy. In the long run, the growth rate of AI enterprises, non-AI enterprises, and total social output will decline.

**$N_t$ remains unchanged, $I_t$ becomes larger, and the labor share $N_t − I_t$ of AI enterprise decreases.** (1) Impact of AI on enterprises in the same industry

Take $\varepsilon = 5$ for simulation. It is initially assumed that the labor force of the two enterprises is heterogeneous and that the production equipment used by the two enterprises is different. The simulation results are shown in Fig 8. When $N_t$ remains unchanged and $I_t$ becomes larger, AI will replace more work tasks. In the same industry, the AI enterprise dominates. Similarly to Fig 1, labor and capital will flow from the non-AI enterprise to the AI enterprise, which will promote industrial transformation and upgrading. Unlike Fig 1, the capital flow is faster. That is, the substitution effect of AI will accelerate the process of industrial structure transformation.

(2) Impact of AI on enterprises in different industries

Take $\varepsilon = 0.5$ for simulation. It is initially assumed that the labor force of the two enterprises is heterogeneous. The simulation results are shown in Fig 9.

When $N_t$ remains unchanged and $I_t$ becomes larger, AI will replace more labor tasks. In different industries, capital and labor will flow from the AI enterprise to the non-AI enterprise. The difference from Fig 9 is that the substitution effect of AI ($I_t$ becomes larger) will lead to a decline in the wages of workers in the AI enterprise. The reason is that the substitution effect

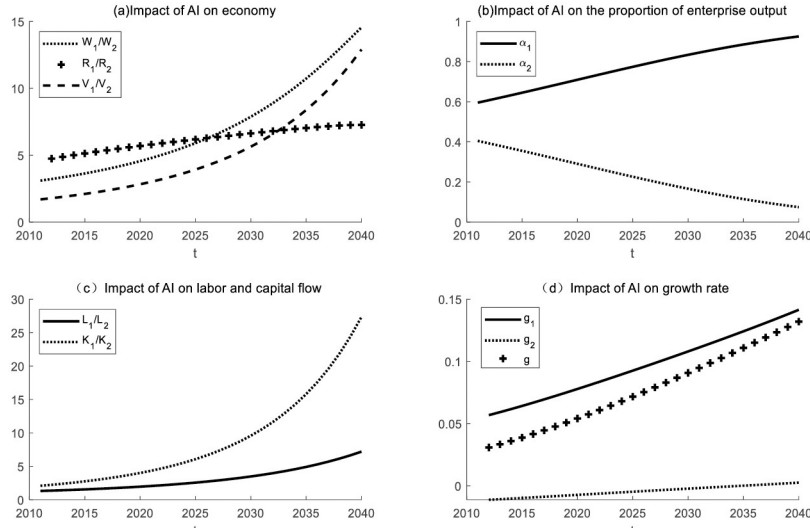

**Fig 8. The impact of AI on enterprises in the same industry with $N_t$ unchanged and $I_t$ increasing.**

of AI is greater than the productivity effect. The improvement of the degree of intelligence ($I_t$ becomes larger) improves the production efficiency, but at the same time leads to the narrowing of the working range $[I_t, N_t]$ of the labor force, which leads to the oversupply of the labor force compared with the previous one, resulting in a decline in wages. At that time, industrial transformation and upgrading fell into a sluggish state. The reason is that the production efficiency of the non-AI enterprise is low, which affects the development of the whole social economy.

**$I_t$ remains unchanged, $N_t$ becomes larger, and the labor share $N_t - I_t$ of AI enterprise increases.** (1) Impact of AI on enterprises in the same industry

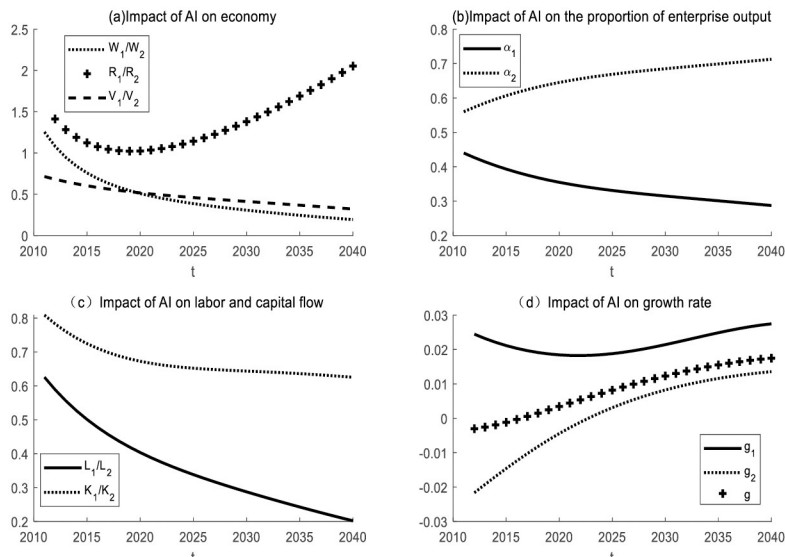

**Fig 9. The impact of AI on enterprises in different industries as $N_t$ remains unchanged and $I_t$ becomes larger.**

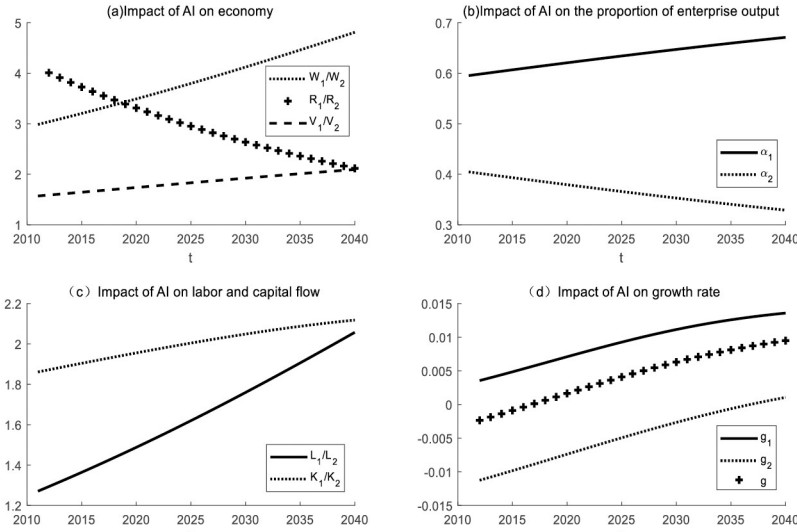

**Fig 10. $I_t$ remains unchanged, $N_t$ becomes larger, and the impact of AI in the same industry.**

Take $\varepsilon = 5$ for simulation. It is initially assumed that the labor force of the two enterprises is heterogeneous and that the production equipment used by the two enterprises is different. The simulation results are shown in Fig 10. The proportion of output of AI enterprise will increase, while that of non-AI enterprise will decrease. Labor and capital will flow from the non-AI enterprise to the AI enterprise. The creation of new tasks will accelerate the flow of the labor force and facilitate the transformation and upgrade of the AI enterprise. The growth rate of total output of AI enterprise, non-AI enterprise and society will increase.

(2) Impact of AI on enterprises in different industries

Take from simulation $\varepsilon = 0.5$, the simulation results are shown in Fig 11. When $I_t$ remains unchanged and $N_t$ becomes larger, the output share of non-AI enterprise 2 increases and that of AI enterprise 1 decreases. Capital will flow from the AI enterprise to the non-AI enterprise. The increase in new tasks will delay the flow of labor. In the first 20 years, the economy will experience negative growth. It took 20 years for the economy to develop soundly.

## Economic impact of subsidy policies adopted by different industries

In order to combine with reality, this part assumes that AI enterprise is manufacturing industries and non-AI enterprise is service industries. To promote the process of industrial intelligence, the government subsidizes enterprises to buy intelligent equipment. The subsidy policy will have two effects, the productivity effect and the consumer effect (as shown in Fig 12). The subsidy policy will reduce the production cost of enterprises and stimulate them to purchase intelligent equipment. With the increase of the subsidy rate, under the effect of the productivity effect, the process of industrial intelligence will gradually improve, and the manufacturing industry will improve production efficiency and output. At the same time, artificial intelligence will lead to a decrease in the labor share of the manufacturing industry. The outflow of labor from the manufacturing industry into the service industry will promote the improvement of the production of the service industry, and ultimately, the total social production will also increase. But at the same time, the intelligence process will be affected by the effect of the consumer. Subsidies will inhibit the development of the service industry and lead to a reduction in

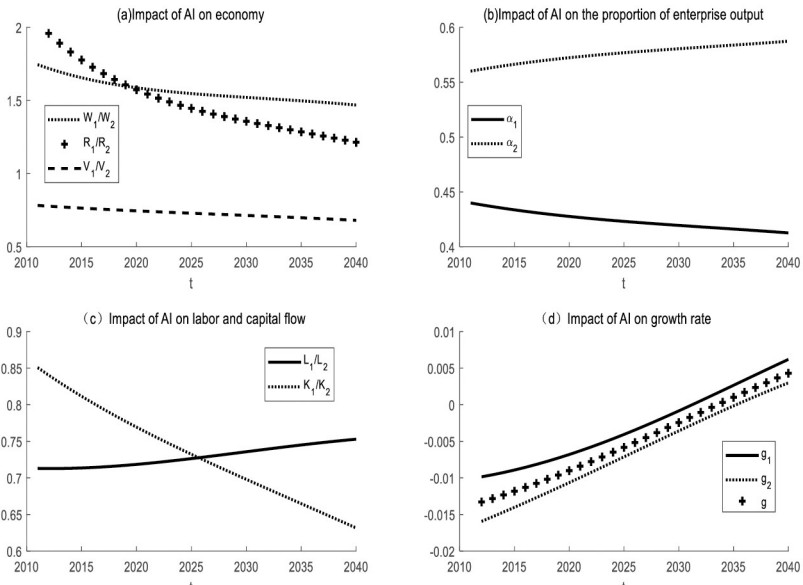

**Fig 11. The impact of AI on different industries with $I_t$ unchanged and $N_t$ increasing.**

the output of the service industry. However, the consumer's consumption attribute is fixed and will not consume more manufacturing products. Inhibited by the consumer effect, the industrial intelligence process shows a gradual downward trend. At this time, the labor share of the manufacturing industry will increase and eventually stabilize. As intelligentization of the manufacturing industry is hindered, production efficiency cannot be steadily improved, resulting in a reduction in manufacturing production and a reduction in total social production. Fig 7 also shows that the productivity effect of the subsidy policy is slowly increasing, while the consumer effect is rapidly declining. Under the combined effect of the two effects, the industrial intelligence process will reach a balance in two years.

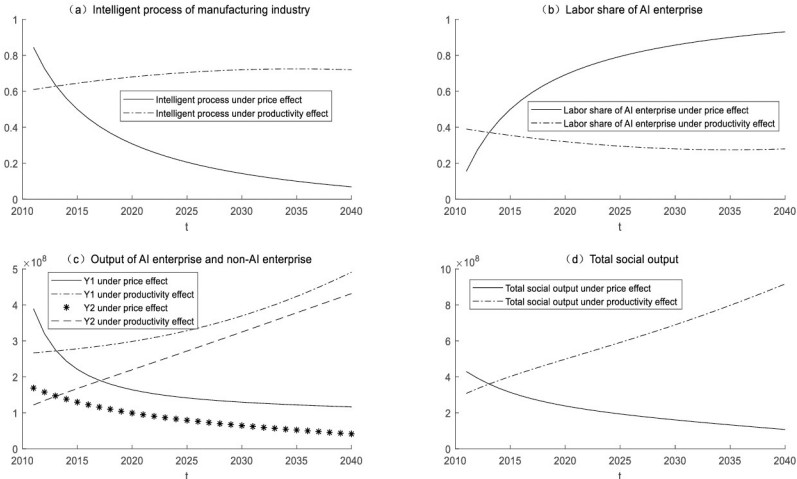

**Fig 12. Economic impact of government subsidies policy.**

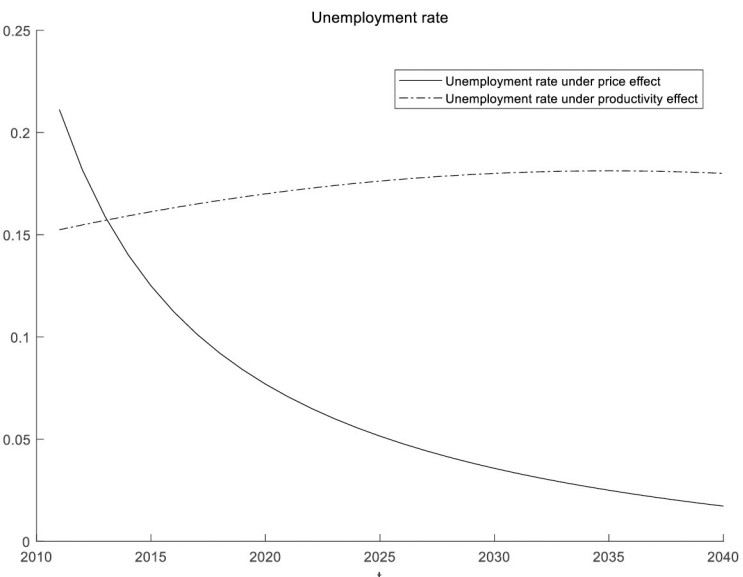

**Fig 13. Unemployment rate under government subsidy policy.**

This article also discusses the unemployment rate under government subsidy policies, as shown in Fig 13. Numerical simulation found that as the subsidy rate increases, enterprises will adopt more intelligent devices. The productivity effect reduces the use of labor and increases unemployment. Consumers have not increased their demand for manufacturing products. Under the suppression of price effects, the intelligentization process of enterprises will stop, so the unemployment rate will not continue to increase.

### Changes in technological measurement

This section will explore the economic impact of enterprises in both industries using artificial intelligence for production through numerical simulation. (as shown in Fig 14).

Enterprise 1 has successfully implemented intelligent production, while Enterprise 2 is gradually integrating artificial intelligence into its production processes. Numerical simulations suggest that Enterprise 2 will gradually catch up with Enterprise 1 in terms of technological advancement and eventually converge with Enterprise 1's level of technological sophistication. Additionally, the production output ratios of Enterprise 1 and Enterprise 2 will tend to approach equilibrium. Initially, Enterprise 1 held a significant advantage in the R&D value. However, as the technological measures of Enterprise 2 continue to advance at a faster pace, the R&D value of Enterprise 2 will experience accelerated growth. In the long run, the economic growth rates of Enterprise 2 and Enterprise 1 are expected to become increasingly similar, contributing to overall economic growth within the broader society.

These findings hold significant real-world implications. Firstly, they signify that in the realm of technology, even with initially significant disparities, Enterprise 2 has the potential to catch up and compete with the leading Enterprise 1. This encourages ongoing investment and innovation in the technology sector, thereby driving technological advancement and economic growth. Secondly, the convergence of production output ratios between Enterprise 1 and Enterprise 2 can reduce resource waste, enhance efficiency, and alleviate the strain on finite resources. This is critical for sustainable development. Most importantly, the increasing

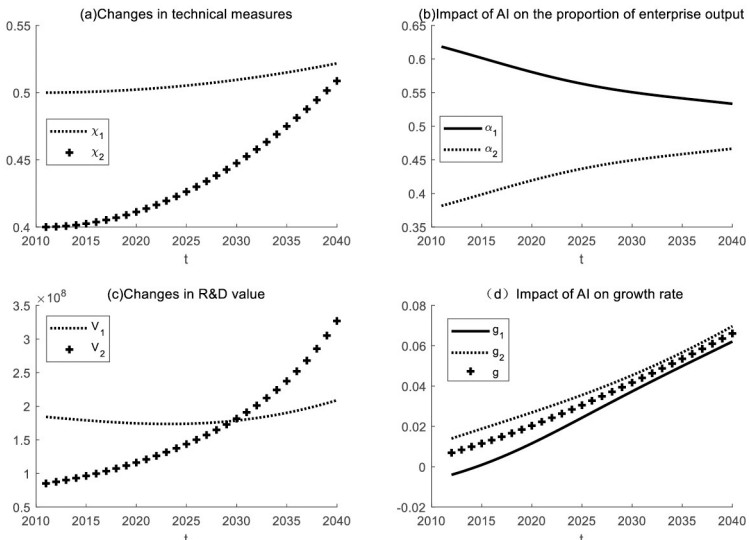

**Fig 14. The economic impact of intelligent production by enterprises in both industries.**

similarity in economic growth rates between Enterprise 1 and Enterprise 2 contributes to reducing internal inequalities within society. This helps ensure that a larger portion of the population shares in the benefits of economic growth, thereby strengthening social stability and sustainability.

In summary, these research findings underscore the potential for technological upgrading and collaboration, promoting economic growth, resource efficiency, and social equity. These changes have far-reaching implications for achieving a more prosperous, fair, and sustainable society.

## Conclusions and recommendations

Artificial intelligence has had a significant impact on enterprise production. AI enables the automation of various production processes, reducing the need for manual intervention. AI-powered robots and machines can handle repetitive tasks with precision and efficiency, leading to increased production speed and accuracy. AI algorithms can analyze data from sensors, cameras, and other sources to detect defects or anomalies in the production process. Machine vision systems equipped with AI can identify and reject faulty products, ensuring consistent quality throughout the production line. AI algorithms can optimize the allocation of resources such as raw materials, energy, and labor. By analyzing production data, AI can identify inefficiencies and bottlenecks in the production process, enabling enterprises to streamline operations, minimize waste, and reduce costs. AI technologies such as collaborative robots can work alongside human workers, enhancing productivity and safety in production environments. AI-powered systems can also provide real-time assistance and guidance to workers, improving overall efficiency and reducing errors.

The impact of artificial intelligence on workers' skills is significant and transformative. While AI technologies can automate certain tasks, they also create new opportunities and challenges for the workforce. AI adoption often requires workers to acquire new skills and knowledge to effectively collaborate with AI systems. Upskilling programs help workers learn how to leverage AI tools, understand data analysis, and develop problem-solving capabilities.

Reskilling initiatives enable workers to transition to new roles that complement AI technologies. AI systems can work alongside human workers, augmenting their capabilities and enabling more efficient collaboration. Workers need to develop skills in working with AI tools, understanding AI-generated insights, and effectively integrating AI into their workflow. AI technologies excel at repetitive tasks, while workers can focus on more complex and creative problem-solving. As AI takes over routine tasks, workers need to develop critical thinking, creativity, and innovation skills to tackle more complex challenges that require human ingenuity.

The uneven development between industries has seriously hindered the process of intelligent manufacturing in the entire society. This article analyzes the impact of AI on industrial transformation and upgrading through theoretical modeling and numerical simulation. In the same industry, AI enterprise will attract capital and labor inflows and gradually occupy the leading position in the industry. AI will eliminate backward production capacity in the same industry and promote the transformation and upgrading of the industry. Between different industries, capital and labor flow from AI enterprise to non-AI enterprise, which will hinder the transformation and upgrading of enterprises. Among different industries, this paper analyzes the policy effect of taxing non-AI enterprise and transferring payments to AI enterprise. The benefit of the subsidy policy is that it can promote the growth of AI enterprise and facilitate the transformation and upgrading of enterprises. However, the disadvantage of the subsidy policy is that it will inhibit the development of non-AI enterprise, which will easily lead to large-scale unemployment in the future. If intelligent production is realized among all sectors of society, the share of capital and labor income of all sectors will gradually tend to the same value when balancing.

The policy implications of this article are as follows.

(i) Strengthen technology research and development to promote the balanced development of artificial intelligence among various industries. Within the same industry, the AI enterprise will occupy the dominant position of the department and will stimulate other companies to accelerate the process of artificial intelligence. However, due to the existence of technological externalities between different industries, the non-AI industry lacks the incentive to innovate. In order to promote the increase of social growth rate, the government can invest funds in the research and development of non-artificial intelligence industries, encourage scientific research institutions to develop technologies that have not realized the automation industry.

(ii) The government can adopt appropriate tax policies to make transfer payments. For example, in the initial stage, in order to stimulate the development of artificial intelligence in all industries, the government can subsidize artificial intelligence enterprises. To help enterprises accelerate the process of artificial intelligence, the government can subsidize enterprises that purchase artificial automation equipment to help enterprises reduce costs.

(iii) Although subsidies can accelerate artificial intelligence, they only have short-term effects. In the long run, in order to promote sustainable and stable economic development, governments and enterprises must adhere to the path of independent research and development.

(iv) The government can offer incentives and rewards to encourage businesses to engage in R&D and innovation in the field of artificial intelligence. These incentives may encompass tax benefits, patent protection, and financial support for R&D, all aimed at stimulating corporate investments in technology and innovation.

(v) The government can foster collaboration and knowledge sharing among different industries to expedite the dissemination and application of technology. This can help lower

barriers to technology diffusion, promoting technology sharing and collaborative innovation.

(vi) The government can implement social welfare policies to alleviate the impact on individuals who lose their jobs due to technological transformations. This may include providing unemployment benefits, retraining opportunities, and career guidance to assist individuals in transitioning smoothly into new employment sectors.

These policy recommendations are designed to promote equitable development of artificial intelligence across diverse industries while ensuring sustainable economic growth and stability in society. Governments, businesses, and research institutions can collaborate to formulate a comprehensive policy framework that addresses the challenges and opportunities of the artificial intelligence era.

While the two-industry model presented in this paper has yielded certain research findings, it still exhibits several limitations, providing fruitful directions for more in-depth investigation in the future:

(i) The paper exclusively considers a closed economic system, neglecting the broader influence of other nations. Future research endeavors can extend this model to open economies to gain a better understanding of the economic implications of disparate artificial intelligence development in developed and developing countries.

(ii) This paper primarily examines the manufacturing and service sectors from a macroeconomic perspective, without delving deeply into the intricacies of these sectors. Future studies could conduct more detailed analyses of specific industries, aligning with real-world scenarios, to unveil additional nuances and insights.

(iii) The focus of this paper on subsidy policies represents only a fraction of public policy considerations. Subsequent research can explore a broader spectrum of public policies, such as increased investments in public education and novel infrastructure, to comprehensively comprehend their impact on the field of artificial intelligence.

## Supporting information

**S1 Appendix.**
(DOCX)

**S1 Data set.**
(ZIP)

## Author Contributions

**Conceptualization:** Xu Huang.

**Formal analysis:** Xu Huang.

**Methodology:** Xu Huang.

**Project administration:** Xu Huang.

**Supervision:** Xu Huang.

**Writing – original draft:** Xu Huang.

**Writing – review & editing:** Xu Huang.

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
