## [Decision Letter · Decision Letter 0]

27 Sep 2023

PONE-D-23-24595Impact of AI on the Transformation and Upgrade of Industrial StructurePLOS ONE

Dear Dr. huang,

Thank you for submitting your manuscript to PLOS ONE. After careful consideration, we feel that it has merit but does not fully meet PLOS ONE’s publication criteria as it currently stands. Therefore, we invite you to submit a revised version of the manuscript that addresses the points raised during the review process.

We look forward to receiving your revised manuscript.

Kind regards,

Donato Morea

Academic Editor

PLOS ONE

Journal Requirements:

2.Thank you for stating the following financial disclosure: 

 "This work was supported by the Ningbo Natural Science Foundation (2023J061), the Ningbo Philosophy and Social Science Foundation(G2023-2-41), Young Innovative Talents of Guang-dong Ordinary Colleges and Universities (Humanities and Social Sciences) (2021WONCX145), the Recycling and Remanufacturing Network Organization and Its Governance under the Sharing Economy of China under Grant (18BGL184), Zhejiang Provincial Natural Science Foundation of China under Grant (LY20G030025)."  

Reviewers' comments:

Reviewer's Responses to Questions

**Comments to the Author**

1. Is the manuscript technically sound, and do the data support the conclusions?

Reviewer #1: Yes

Reviewer #2: Partly

2. Has the statistical analysis been performed appropriately and rigorously? 

Reviewer #1: I Don't Know

Reviewer #2: N/A

3. Have the authors made all data underlying the findings in their manuscript fully available?

Reviewer #1: Yes

Reviewer #2: No

4. Is the manuscript presented in an intelligible fashion and written in standard English?

Reviewer #1: Yes

Reviewer #2: No

5. Review Comments to the Author

Reviewer #1: The paper presents a two-industry model that examines the dynamics between AI and non-AI enterprises. While the topic is timely and holds significant relevance to the current trends in the global economy, there are various aspects of the manuscript that require revision and clarification. Below are the suggestions:

Clarity of Model Assumptions:

The assumptions underlying your two-industry model are not clearly laid out in the abstract. The model's soundness can only be ascertained with a clear understanding of these assumptions. Kindly elucidate.

Methodological Approach:

The methodology used to derive conclusions from the model should be detailed in the main body. For instance, how are you quantifying the 'consumption effect'? How is the flow of labor and capital between enterprises measured or predicted?

Taxation and Policy Implications:

The abstract suggests a governmental intervention via taxation to promote the improvement of production efficiency. However, such a recommendation demands a deeper empirical or simulation-based analysis. The consequences of such taxation, especially in the context of potential unemployment, need a rigorous analysis.

Clarification on 'Consumption Effect':

The term 'consumption effect' is introduced without a clear definition or context. Please explain the mechanism behind this effect, especially in the context of your model.

Relevance of 'Labor and Capital Share':

The conclusion mentions the tendency of labor and capital share to converge. While this is an interesting result, its relevance and implications in the broader context of the paper should be clarified.

Data Source and Real-world Applications:

The model would greatly benefit from real-world data or case studies that support your findings. If the conclusions are purely theoretical, the paper should also provide suggestions on how to test these conclusions empirically.

Reviewer #2: Title and Scope:

Consider refining the title for a clearer indication of the primary focus and outcomes of the study. Perhaps: "Dynamics of Labor and Capital in AI vs. Non-AI Industries: A Two-Industry Model Analysis".

Elaborate on the significance and implications of the identified imbalance in AI development across industries in the introduction.

Model Assumptions:

Clearly state and justify the underlying assumptions for your two-industry model. How are "AI enterprises" and "non-AI enterprises" explicitly defined?

Discuss the real-world relevance and validity of these assumptions. Which industries or scenarios might this model be most applicable to?

Model Dynamics:

Provide a detailed explanation and illustration (perhaps a flowchart or diagram) of how labor and capital move between the AI and non-AI enterprise under different conditions.

Clarify the "consumption effect" that benefits the non-AI enterprise more. Is this a well-established concept, or is it novel to this paper? Please provide adequate references or theoretical underpinnings.

Government Intervention:

Discuss the rationale and potential consequences of taxing non-AI enterprises while subsidizing AI enterprises. Would this approach be feasible in practice?

Delve deeper into the ethical considerations of such policies, especially when considering their potential to exacerbate unemployment and further inhibit non-AI enterprise development.

Short-Term Effects:

The paper mentions that taxation has only a short-term effect on AI development. Please provide evidence or theoretical reasoning behind this claim.

Consider examining the potential long-term impacts of these policy suggestions, both positive and negative.

Unemployment Concerns:

Address the unemployment risks in more detail. What industries or worker profiles might be most vulnerable? How significant is the risk?

Equilibrium Analysis:

The conclusion suggests that labor and capital shares will equalize when both industries use AI. Provide a thorough analysis of how and why this equilibrium is reached.

Discuss the societal and economic implications of such an equilibrium. What does it mean for industry growth, innovation, and wealth distribution?

Literature Review:

Ensure that the paper incorporates a comprehensive review of existing literature, touching upon AI adoption rates across industries, economic implications of AI, and policy recommendations from other studies.

How does this study differentiate or expand upon existing research?

Limitations:

Clearly state the limitations of the two-industry model. Are there external factors or industry-specific nuances that the model doesn't account for?

Recommendations:

Beyond taxation, offer alternative policy recommendations or corporate strategies that could alleviate the identified challenges or optimize the benefits of AI adoption across industries.

6. PLOS authors have the option to publish the peer review history of their article (what does this mean?). If published, this will include your full peer review and any attached files.

Reviewer #1: No

Reviewer #2: No

---

## [Author Response · Author response to Decision Letter 0]

27 Oct 2023

There are many comments to respond to reviewers, please see Response to Reviewers.doc.

---

## [Decision Letter · Decision Letter 1]

16 Nov 2023

Dynamics of labor and capital in AI vs. Non-AI industries: A two-industry model analysis

PONE-D-23-24595R1

Dear Dr. huang,

We’re pleased to inform you that your manuscript has been judged scientifically suitable for publication and will be formally accepted for publication once it meets all outstanding technical requirements.

Best regards,

Prof. (Assist.) Donato Morea, Ph.D.

Academic Editor

PLOS ONE

Reviewers' comments:

Reviewer's Responses to Questions

**Comments to the Author**

1. If the authors have adequately addressed your comments raised in a previous round of review and you feel that this manuscript is now acceptable for publication, you may indicate that here to bypass the “Comments to the Author” section, enter your conflict of interest statement in the “Confidential to Editor” section, and submit your "Accept" recommendation.

Reviewer #1: All comments have been addressed

Reviewer #2: All comments have been addressed

2. Is the manuscript technically sound, and do the data support the conclusions?

Reviewer #1: Partly

Reviewer #2: Yes

3. Has the statistical analysis been performed appropriately and rigorously? 

Reviewer #1: Yes

Reviewer #2: I Don't Know

4. Have the authors made all data underlying the findings in their manuscript fully available?

Reviewer #1: No

Reviewer #2: Yes

5. Is the manuscript presented in an intelligible fashion and written in standard English?

Reviewer #1: No

Reviewer #2: Yes

6. Review Comments to the Author

Reviewer #1: The author has made revisions to the article in accordance with my suggestions. I recommend acceptance.

Reviewer #2: I appreciate for the authors' revision. The existing paper has original content and worthy for publication in the journal. I can recommend it for a possible publication.

7. PLOS authors have the option to publish the peer review history of their article (what does this mean?). If published, this will include your full peer review and any attached files.

Reviewer #1: No

Reviewer #2: No

---

## [Editor Report · Acceptance letter]

29 Nov 2023

PONE-D-23-24595R1 

Dynamics of labor and capital in AI vs. Non-AI industries: A two-industry model analysis 

Dear Dr. Huang:

I'm pleased to inform you that your manuscript has been deemed suitable for publication in PLOS ONE. Congratulations! Your manuscript is now with our production department. 

Kind regards, 

on behalf of

Professor (Assistant) Donato Morea 

Academic Editor

PLOS ONE